# Effectiveness of Residential Treatment for Juveniles with Problematic Sexual Behavior: A Systematic Review

**DOI:** 10.3390/ijerph192315625

**Published:** 2022-11-24

**Authors:** Whitney Howey, Brad Lundahl, Andrea Assadollahi

**Affiliations:** College of Social Work, University of Utah, Salt Lake City, UT 84112, USA

**Keywords:** residential treatment, juvenile, sex offense, effectiveness, recidivism

## Abstract

Juveniles with problematic sexual behaviors are often placed in residential treatment. However, little is known about the effectiveness of such treatments in terms of reducing recidivism or enhancing mental wellbeing. To better understand the impact of residential treatment for these juveniles we conducted a Systematic Review on studies that reported recidivism rates. PRISMA guidelines were followed. 1126 studies were initially identified, with only six meeting the final inclusion criteria. Sexual recidivism rates averaged 5.20% across the six studies, which is similar rates of non-residential treatments. The results suggest that certain client factors predict recidivism, for example youth labeled as obsessive offenders were more likely to recidivate compared to those labeled as opportunistic. Most studies also measured non-sexual crimes post treatment; recidivism rates for sexual misconduct tended to be lower than for other crimes. Despite the significant intrusion of residential treatment centers, remarkably few empirical studies exist to establish their effectiveness in reducing recidivism. The comparable recidivism rates to non-residential treatment programs begs the question of whether residential centers add value beyond outpatient care and suggest that less restrictive interventions may be sufficient. Helping youth evidencing problematic sexual behaviors involves complex dynamics, however caution is recommended on relying on residential treatment.

## 1. Introduction

Despite limited evaluation research, residential treatment centers are a commonly relied upon intervention for juveniles who have sexually offended (hereafter: juveniles with problematic sexual behavior) (Shapiro et al., 2002) [1]. In 1992, over 750 outpatient and residential treatment programs operated for juveniles with problematic sexual behaviors in North America (Burton & Smith-Darden, 2000) [2]. By 2000 the number of programs declined by 61% to 291, with only 101 residential and 190 community-based (non-residential) programs (Burton & Smith-Darden 2001) [3]. This significant decline suggests a change in the treatment of juveniles with problematic sexual behaviors. A 2009 survey (McGrath et al., 2010) [4] identified 373 adolescent treatment programs for problematic sexual behaviors, with approximately 18% (*n* = 98) being residential programs—representing a further decline. Burton and Smith-Darden (2000) [2] suggested that the decrease over time in these treatment programs results from beliefs about the ineffectiveness of treatment and little programmatic outcome data to contradict this belief.

At the time of this writing, residential treatment centers are still being operated despite uncertainties over their effectiveness. When youth sexually offend a variety of interventions are available to promote rehabilitation and reduce the risk of re-offending, including outpatient care, intensive outpatient care (IOP), and residential treatment. Compared to outpatient treatment, residential treatment is resource-intensive, removes youth from their home, and may actually promote criminal behavior (Hunter et al., 2004) [5]. In terms of financial costs, residential treatment for one juvenile with problematic sexual behaviors in the United States costs approximately $130,000 per year—often funded by tax dollars (Pratt, 2013) [6].

Certainly, helping youth who sexually offend involves complex dynamics such as the wellbeing of the youth, developmental trajectories, schooling, and, potentially, the continued safety of victims who may or may not live under the same roof (see Pratt, 2013) [6]. There are circumstances when restrictive settings for a youth who offends might be explored and warranted. Bourke and Donohue (1996) [7] recommended residential treatment when the following conditions are present: numerous sexual offenses, numerous victims, aggression accompanying the offenses, presence of antisocial attitudes, emotional and behavioral problems, lack of motivation for treatment, unsafe home environment, and if the victim resides in the same home.

When a youth who has offended needs to be removed from their home to protect a victim a variety of factors should be considered to promote effective care. Considering individual differences such as varying levels of risk to reoffend (Hunter et al., 2004) [5] and the youth’s relative willingness and ability to engage in treatment (Pratt, 2013) [6] have been shown to be important. Research suggests that placing a youth who is open to treatment and a lower risk for reoffending profile in a residential treatment program with offenders who have a more serious risk history may actually increase their risk to reoffend (Hunter et al., 2004) [5].

There is a building consensus that juvenile sexual offenders are generally at low risk for sexual recidivism, especially when compared to general or ‘nonsexual’ recidivism (Kettrey & Lipsey, 2018) [8]. For example, Hunter and colleagues (2004) [5] found that among juvenile sexual offenders who attended a wraparound community-based treatment only 2% sexually recidivated while 23% recidivated non-sexually a year after treatment. Another study examining an outpatient treatment program found a 1.63% sexual recidivism rate and 9.5% non-sexual recidivism rate after two years (Kolko, 2006) [9]. Seabloom et al. (2003) [10] evaluated an intensive outpatient program for sexually offending youth and reported no recidivism (i.e., 0%) among those who completed the program. Interestingly, they also concluded that clients living with their parents were more likely to complete the program, which should lower their risk for sexual recidivism (Seabloom et al., 2003) [10]. We identified three systematic reviews that examined the effectiveness of treatment for youth who have sexually offended (Kettrey & Lipsey, 2018; Reitzel & Carbonell, 2006; Walker et al., 2004) [8,11,12]. Despite using broad search strategies, none of these identified more than 10 studies. Additionally, across the three systematic reviews only four individual studies included residential treatment centers with the majority being some form of outpatient treatment. Effect sizes ranged considerably across the studies, with some recidivism rates being in small range and others in the large range.

The lack of reviews on residential treatment placements for youth who have sexually offended leaves a gap in knowledge in terms of effectiveness and possible moderators. Our study seeks to respond to this gap in the literature. Specifically, the present study is a systematic review of research exploring the effectiveness of residential treatment facilities for juveniles with problematic sexual behaviors using recidivism rates as the indicator of effectiveness. Additionally, we explored whether certain aspects of a treatment program would moderate outcomes. Two questions guided our review. First, how effective are residential treatment programs for juveniles with problematic sexual behaviors as measured by recidivism rates? Second, what elements of residential treatment programs contribute to lower recidivism rates?

## 2. Methods

The preferred reporting items for systematic reviews and meta-analyses (PRISMA, 2020; Page et al., 2020) [13] guidelines were followed throughout the process.

### 2.1. Search Strategy

To be included, studies needed to (a) directly examine the effectiveness of residential treatment centers for juveniles with problematic sexual behaviors, (b) provide data on recidivism, (c) focus on youth from the age of 11–22 years of age (many treatment centers extend the age bracket for juveniles to 22 years), (d) be published in a peer-reviewed journal, and (e) be published in English. Studies were excluded if they (a) focused on correctional facilities, detention facilities, and day-treatment programs or community-based programs where youth lived at home or in the foster care system, (b) focused on adult populations, and (c) were published before 2000.

In an effort to identify all relevant studies, the following twenty databases were searched: Academic Search Premier, Academic Search Ultimate, Alt HealthWatch, APA PsycArticles, APA PsycInfo, Criminal Justice Abstracts, ERIC, Family & Society Studies Worldwide, Health and Psychosocial Instruments, Health Source—Consumer Edition, Health Source: Nursing/Academic Edition, Humanities & Social Sciences Index Retrospective: 1907–1984 (H.W. Wilson), Legal Collection, Legal Information Reference Center, Library, Information Science & Technology Abstracts, MEDLINE, MEDLINE Complete, Professional Development Collection, Psychology, and Behavioral Sciences Collection, and Social Work Abstracts.

The following broad search terms were used: (adolescents or teenagers or young adults or teen or youth) OR juvenile AND residential treatment OR (treatment or intervention or therapy) AND (sexual offenders or sex offenders or sexual abusers) OR sexual perpetrator. While measurement of recidivism rate is a key inclusion criterion, we did not use this in the search criteria to allow for a broad return of studies which could then be narrowed through coding each study. This search was performed on December 8, 2020 and terminated on 16 January 2021. Only English full-text resources were accepted for this review.

### 2.2. Coding Strategy

We utilized a specialized systematic review software designed to organize and screen articles (Covidence, 2021) [14]. Two independent reviewers screened articles for inclusion. Any disagreements among reviewers were discussed and a third reviewer was utilized, if necessary.

### 2.3. Synthesis Methods

Using a code sheet developed for this study, two coders extracted information on study characteristics, study design characteristics (e.g., methodological rigor), intervention characteristics, participant characteristics, and outcomes. Study rigor was assessed using criteria from existing assessment instruments and approaches such as the Cochrane system (Cooper et al., 2009; Higgins & Green, 2011; Lipsey & Wilson, 2000) [15,16,17]. Each study was rated on a 14-point scale that examined criteria such as number of participants, attrition, quality control, objectivity of measurements, and reporting of follow-up data (Rigor Rating code sheet is available upon request). The National Institute of Health (NIH, 2022) [18] rates the overall quality based on the following guidelines: 0–4 = Poor, 5–10 = Fair, and 11–14 = Good. Outcome measurements (i.e., recidivism) included type of recidivism reported (i.e., general, sexual, or both), how recidivism was measured (i.e., record review, self-report), recidivism follow-up (in years or time points), and recidivism rates (in percentages). Characteristics of interventions included average length of stay (in days), intervention type (i.e., cognitive behavioral therapy, mode deactivation therapy), intervention delivery (i.e., individual psychotherapy, group therapy, family therapy), who delivered the intervention (i.e., licensed mental health counselor/social worker, psychologist, staff), amount of intervention received by clients per week, additional services provided by residential treatment facility (i.e., school, sports, recreation, medication management, family visits/contact), client classification into groups (i.e., child offenders, peer offenders, obsessive, opportunistic), and successful completion rates. Characteristics of study samples included the location of study, sample size, gender, average age of clients (in years), and racial/ethnic background. Studies varied in how much information they provided. 

### 2.4. Coding Reliability

Coding reliability was examined at two stages, article selection and evaluating study characteristics. At the study selection level, Cohen’s Kappa was 0.61, which is considered moderate (McHugh, 2012) [19]. Of note, the relatively low rate of agreement seemed to stem from one reviewer, including studies that seemed theoretically relevant despite not meeting the inclusion criteria. Interrater reliability was higher for evaluating study characteristics, with proportionate agreement values averaging 0.93. All differences between coders were resolved by further discussion, involving a third reviewer if necessary.

## 3. Results

### 3.1. Study Selection

A total of 1264 documents were initially identified, which reduced to 1126 after duplicates were removed. As shown in Figure 1, most studies were excluded at the title and abstract review stage because they focused on adult populations or were not residential treatment facilities (*n* = 987). The remaining 139 studies, which passed the title and abstract review, were submitted to a full-text review. The majority of the articles (*n* = 133) were excluded because they did not include recidivism data or were not residential treatment centers. Of the 139 studies, only 6 met our full inclusion criteria. All identified studies were able to be fully secured.

### 3.2. Study Characteristics

The final sample for the review consisted of six studies. See Table 1 for a brief overview of the included studies, as well as the study quality score located below the study year. Study characteristics, participant characteristics, intervention characteristics, and recidivism outcomes of the studies are summarized below.

Each study was assessed for rigor and quality. All the studies received a quality rating of Fair. All but one study, Hendriks and Bijleveld’s (2008) [20], was conducted in the in the United States. The total number of participants across all six studies was 926 (*n* = 926). Some participants resided in a residential treatment center but were non-sex offenders, *n* = 126 (14%). In one study, these non-sex offenders were either substance-using or general offenders compared to juveniles with problematic sexual behaviors.

Of interest, some studies attempted to better understand outcomes by exploring possible moderators. For example, Kemper and Kistner (2007) [21] used victim type to classify participants as either child, peer, or mixed offenders. Another study, Hendriks and Bijleveld (2008) [20] used perpetrator type to classify participants as either obsessive (67%) or opportunistic (33%) offenders. Gillis and Gass (2010) [22] compared their residential treatment program (LEGACY) to “Other Specialized Programs” and a “Youth Detention Center,” (Note: Only the results of the LEGACY program are of interest for this review).

Four of the studies reported on the average length of stay which was approximately 15 months (435 days, SD = 88) [1,21,22,23,24].

### 3.3. Participant Characteristics

All participants were male. The average age was 15 years (SD = 1.5). Of the studies that reported on ethnicities, white participants were most common at 55.5% (SD = 27.7), black participants were next at 42.3% (SD = 25.3), and Native Americans were <1% (SD = 0.01), biracial were 1.2% (SD = 2.4), and “other” were <1% (SD = 1.7).

**Table 1 ijerph-19-15625-t001:** Brief overview of included studies.

Study	Purpose of Study	Measures & Methods	Analysis	Participant Characteristics	Key Findings
Calleja (2015) [23] [Study Quality = 6]	Examine comparative recidivism between subtypes of adolescent offenders and to investigate four commonly perceived risk factors related to adolescents who have sexually offended	Comparison between general, substance using and sexual offenders. Recidivism evaluated at 1 year post release from residential treatment Database search for recidivism up to 2 years post release	Logistic regression with stepwise and backward variable selection with four explanatory variables	*n* = 166 sexual offender *n* = 40 100% Male avg. length of stay (JSO) = 462.09 days 16.3% White 78.3% Black 0.01% American Indian 4.8% Biracial	Only 3% of juvenile sexual offenders reoffended (general) Zero sexual recidivism
Gillis & Gass (2010) [22] [Study Quality = 10]	Examine effectiveness of the LEGACY behavior Management model in aggregate using adventure programming	Matched group design compared to two other programs with similar juveniles, within the same state and time period Recidivism Archival data: 1989–2002	Chi-square on re-arrest or no re-arrest data a 1, 2, & 3 years Effect sizes ANOVA Survival function Probability of re-arrest	*n* = 285 *n* = 95 in RTC (LEGACY) *n* = 95 OSP *n* = 95 YDC Avg. age = 13.75 100% Male 65.3% White 34.7% Black	LEGACY program had overall less re-arrest rates Significant differences between days from release until re-arrest for LEGACY
Hendriks & Bijleveld (2008) [20] [Study Quality = 8]	Investigate recidivism among juvenile sex offenders who had been treated in a residential setting	Screened files examining recidivism, background personality, environmental criminal career, offense treatment variables, Juvenile Sex Offender Checklist JSOAP, ERASOR, ABV-K, ATL, NPV-J, and WISC-RN	Survival Analyses Descriptive statistics Survival Models	*n* = 114 Avg. age = 16 Opportunistic = 38 Obsessive = 76	11% sexually recidivated 70% re-offended to any offense of Treatment had no relationship with recidivism risk
Kemper & Kistner (2007) [21] [Study Quality = 6]	Compare mixed offenders to other victim age-based subgroups (child and peer offenders), examine sexual and nonsexual histories, and compare groups on important outcomes: treatment performance and recidivism	Archival information Examined sex offense details (number of charges and victims, age at time of offenses, and victim info), criminal history, treatment performance/completion, and recidivism (legal database)	ANOVA Chi-square ANCOVA Survival analysis Log-rank Mantel-Cox	*n* = 296 100% Male Avg. age = 16.01 59.5% White 37.2% Black 66.9% child offenders 26.0% peer offenders 7.1% mixed offenders	Over 40% recidivated, 6% sexually Mixed offenders less likely to successfully complete treatment
Shapiro, Welker &Pierce (2002) [1] [Study Quality = 7]	Investigate the effectiveness of a residential treatment program for boys with histories of sexually aggressive behavior	Achenbach Measures Child Behavior Checklist The Jessness Inventory Adolescent Cognitions Scale Target Behavior Rating Scale Critical Incidents Process Measures Recidivism	T-tests Missing data interfered with some analyses	*n* = 26 100% Male Avg. age = 13.07 81% White 19% Black	10 of 12 measures showed improved functioning No adjudications in follow-up period, though credible allegations made
Thoder & Cautilli (2011) [25] [Study Quality = 6]	Evaluate if Mode Deactivation Therapy (MDT) is more effective than treatment as usual (TAU) with juvenile offenders	Child Behavior Checklist Youth Self-Report Devereux Scales of Mental Disorders Fear Assessment Beliefs Analysis of Aggression, Victims, Intimacy, and Control JSOP-A Reading test Recidivism	Score differences between pre and post tests on all measures	*n* = 39 100% Male 14–17 years old	Results support MDT in treatment for this population 7% overall recidivism 0% sexual recidivism

### 3.4. Intervention Characteristics

The intervention methods included Cognitive Behavioral Therapy only, Mode Deactivation Therapy only, Adventure Based intervention only, a combination of Cognitive Behavioral Therapy and Psychoeducation, and a mixture of interventions such as social skills, aggression/regulation therapy, creative therapy, and music therapy. Overall, individual and group therapy was the primary intervention delivery method and delivered by a licensed mental health professional. Few of the studies reported on the amount of interventions youth received, precluding the ability to examine a possible dose–response analysis. Calleja (2015) [23], Hendriks and Bijleveld (2008) [20], and Kemper and Kistner (2007) [21] reported on residents’ engagement or successful completion rates. Calleja (2015) [23] considered treatment completion as a risk factor for increased recidivism and used the youth’s treatment program completion status as part of data collection and analysis. Hendriks and Bijleveld’s (2008) [20] study reports that 75% of their participants successfully completed treatment. Kemper and Kistner (2007) [21] reports client engagement in group therapy and treatment non-completers. Interestingly, their study tracked when and why clients were removed from group therapy and applied this element of treatment engagement in their overall interpretation of treatment success, or lack thereof (Kemper & Kistner, 2007) [21].

While schooling and other activities were occasionally discussed as part of the residential treatment program, it was impossible to ascertain how residents generally spent their time in terms of treatment, daily living, and other activities. Finally, little is known about family visit patterns, with only one study reporting that they facilitated family visits or family contact (Thoder & Cautilli, 2011) [25] and one other mentioning it being “ideal” (Shapiro et al., 2002) [1].

In sum, treatment variables were of particular interest to this research team, yet the available data do not permit for more refined, quantitative analysis of possible moderators. Table 2 provides a visual of the six studies and their report of intervention specifications. For example, if a study clearly specified intervention type, it was marked as present (X). If a study did not report on the amount or dose of intervention the youth received each week while in treatment, it was marked as not reported (NR).

### 3.5. Recidivism Outcomes

All studies reported on sexual recidivism rates from record reviews, specifically court or police reports. Some studies also reported on non-sexual or “general” recidivism. Table 3 summarizes the six included studies, including the percentage of youth who had sexually offended, recidivism rates, time beyond treatment when recidivism was assessed, and, for some studies, comparisons, or moderators on recidivism.

Notably, all six studies defined recidivism slightly different. Calleja (2015) [23] defined recidivism as a new adjudication, not just a rearrest (consistent with previously established parameters) and required disposition in the juvenile or adult criminal justice systems. Gillis and Gass (2010) [22] explained recidivism as a “rearrest” and further defined rearrest as committing a re-offense and receiving a disposition after release, including technical violations and status offenses. Hendriks and Bijleveld (2008) [20] defined recidivism as a new conviction as evidenced by charge sheets or “judicial documentation.” Kemper and Kistner (2007) [21] vaguely defined recidivism as arrests and convictions in the adult correctional system only and did not account for arrests or convictions in the juvenile system following release. Shapiro and colleagues (2002) [1] broadly defined recidivism as a new adjudication of either sexual or non-sexual offenses. Finally, Thoder and Cautilli (2011) [25] did not clearly define their parameters for recidivism and simply stated “no felony arrests” or “had criminal charges.”

Denote that Gillis and Gass’ (2010) [22] comparison study included an OSP (Other Specialized Programs) and a YDC (Youth Detention Center) group. The sexual and general recidivism data from these two comparison groups were not included in our overall recidivism results because they were not explicitly identified as residential treatment centers.

The sexual recidivism rates ranged from 0% to 11%, with an overall average of 3.86%. Though Shapiro, Welker, and Pierce’s study did not result in any adjudications in their one-year follow-up, there were credible allegations made and considered true by the author. By including what the author considers “true allegations” and therefore sexual recidivism, this increased the average sexual recidivism rate to 5.20%.

The general recidivism rates ranged from 0% to 50%, with an overall average of 19.39%. Again, by including considerations of “true allegations,” the general recidivism rate increased to 23.89%. Follow-up periods ranged from 1-year post-release to 5.2 years. The average follow-up period for sexual and general recidivism was three years.

## 4. Discussion

Our ability to answer the question of how effective are residential treatment centers in helping youth who have sexually offended is limited because only six studies were identified. Across all studies, recidivism rates were rather low, in the 5.20% range. With the exception of Gillis and Gass (2010) [22], none of the studies compared their treatment success against other forms programs or utilized control groups, leaving open the question of whether recidivism rates could be attributed to the intervention or other variables. Our results suggest that recidivism rates do not differ much between residential and non-residential, less-restrictive programs. Across all studies, there appears to be clear evidence that recidivism rates are far smaller for sexual behaviors compared to general recidivism.

Two studies reported on moderator characteristics linked to sexual recidivism for youth [20,21,26]. Hendriks and Bijleveld (2008) [20] found that recidivism rates were much higher, over 500%, for youth classified as having “obsessive offenders” compared to those who’s offense appeared to be opportunistic. “Obsessive offenders” often abused younger children, averaging 5 or more years younger than themselves and also exhibit more dominant sexual motives (Hendriks & Bijleveld, 2008) [20]. Kemper and Kistner’s (2007) [21] work found that youth who sexually offended and had targeted children, versus same aged peers, were far more likely to recidivate. Specifically, of the 18 youth in their study who sexually recidivated after treatment a full 17 had targeted a young child. While it is helpful to know that youth characteristics predict outcomes, our study found no evidence that interventions were tailored to the youth based on such characteristics. Furthermore, there seems to be no tailoring of interventions based on the juvenile’s age, cultural considerations, language, or developmental levels. There is no demonstratable evidence that residential treatment is needed for particular groups (i.e., opportunistic and peer). Mixed offenders are an example from this review, considering the treatment for this group compared to peer offenders did not seem to differ (Kemper & Kistner, 2007) [21]. Yet, the findings show significant differences, suggesting that the driving force to recidivate or not is something inside the youth versus the result of treatment.

Recidivism rates from residential treatment centers appear comparable to rates from community-based treatment centers challenging the logic of using residential treatment centers over less-restrictive programs (Seabloom et al., 2003) [10]. In this vein, none of the included studies offered a theoretical rationale for preferring residential treatment placement over a less-restrictive placements. Furthermore, given that none of the study designs directly compared residential treatment centers with non-residential treatments it is unknown if recidivism rates are linked to the intervention or passage of time.

The findings of this systematic review must be interpreted within the boundaries of its limitations. No study directly compared outcomes from residential and non-residential settings, leaving open questions about the relative effectiveness of each. Further, our study only focused on recidivism rates versus other outcome indicators such as mental health or thinking patterns which may be more proximal than recidivism. While we hoped to statistically combine (i.e., meta-analyze) outcomes, there were too few studies to have confidence in such combinations and only one reported a head-to-head comparison of residential treatments to other interventions. Additionally, due to the inability to do cross-study synthesis, all bias is within each study. In this vein, our study could not quantitatively identify possible moderators to intervention beyond directly reporting what primary studies offered. Thus, our review offers no guidelines on how long treatment should be, if certain intervention characteristics should be pursued or avoided, or if treatments should be modified based on characteristics of the youth despite clear data that some youth characteristics are very important predictors of recidivism (victim age; obsessive versus opportunistic offending). Lastly, from the six studies we identified, there is little information as to what residential treatment really entails. Details are absent about the dosage, frequency, and interventions being used in residential interventions. To obtain more information and a better understanding of what residential treatment entails, a scoping review may be plausible for future research.

Our review is the first known study focused specifically on residential treatment facilities for juveniles with problematic sexual behaviors. The results suggest that residential treatment for youth who have sexually offended may not be necessary given the relative effectiveness of non-residential programs (Hunter et al., 2004) [5]. Further, no study assessed for possible problems that might result from prolonged residential treatment away from caretakers.

## 5. Conclusions

Residential treatment facilities are still being used for youth who have sexually offended. This review highlights the extant literature regarding the effectiveness of residential treatment facilities in reducing recidivism for this population. Our findings reveal comparable recidivism rates of residential treatment to non-residential treatment programs, suggesting that less restrictive interventions such as community-based programming may be sufficient. Helping youth evidencing problematic sexual behaviors involves complex dynamics; however, caution is recommended on relying on residential treatment.

## Figures and Tables

**Figure 1 ijerph-19-15625-f001:**
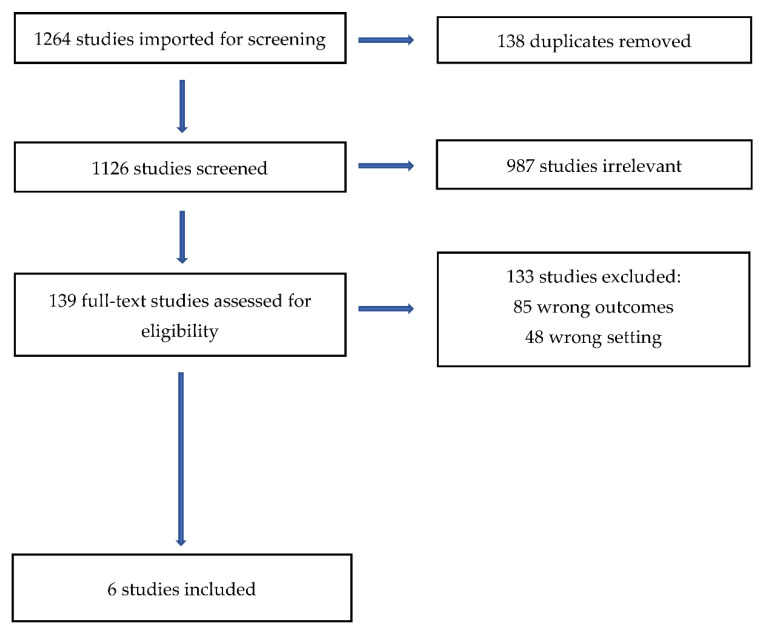
Covidence PRISMA.

**Table 2 ijerph-19-15625-t002:** Intervention specifications of included studies.

Study	Intervention Specifications	Dose of Intervention	Treatment Performance	Activities Offered (School, Sports, Other)	Family Involvement
Calleja (2015) [23]	X	NR	X	NR	NR
Gillis & Gass (2010) [22]	X	NR	NR	X	NR
Hendriks & Bijleveld (2008) [20]	X	NR	X	X	NR
Kemper & Kistner (2007) [21]	X	NR	X	NR	NR
Shapiro, Welker, & Pierce (2002) [1]	X	X	NR	X	X
Thoder & Cautilli (2011) [25]	X	X	NR	NR	X

**Table 3 ijerph-19-15625-t003:** Overview of Sample & Outcomes.

Study	% of Sample Who Are Sexual Offenders	Sexual Recidivism (Follow-up Time)	General Recidivism (Follow-up Time)
Calleja (2015) [23]	24%, (*n* = 40) *n* = 166 32 (18.5%) Substance-using 101 (58.4%) General 40 (23.1%) Sexual	0% (2 years)	Overall: 23.4% Substance-using: 19% General: 32.9% * Sexual: 3% (2 years)
Gillis & Gass (2010) [22]	NR: 100% implied based on placement *n* = 285 *n* = 95 in RTC (LEGACY) *n* = 95 OSP *n* = 95 YDC	* LEGACY = 5.3% OSP = 8.4% YDC = 5.3% (3 years)	*** LEGACY = 13.7% OSP = 24.2% YDC = 29.5% (3 years)
Hendriks & Bijleveld (2008) [20]	100% *n* = 114 76 (66.6%) Obsessive 38 (33.3%) Opportunistic	Overall: 11.4% Obsessive: 9.6% Opportunistic: 1.8% (3 years)	50% Classification group differences not reported (3 years)
Kemper & Kistner (2007) [21]	100% *n* = 293 With reported outcomes 198 (66.9%) Child offenders 77 (26.0%) Peer offenders 21 (7.1%) Mixed offenders	Overall: 6.48% Child: 8.16% Peer: 1.32% Mixed: 4.76% 94% had a prior child victim (17/18 either child or mixed offender) (5.2 years)	Overall: 42.66% Child: 41.33% Peer: 46.05% Mixed: 42.86% (5.2 years)
Shapiro, Welker, & Pierce (2002) [1]	100% *n* = 26	0% 8% considered true ^a^ (1 year)	0% 27% considered true ^a^ (1 year)
Thoder & Cautilli (2011) [25]	100% *n* = 39	0% (4 years)	7% (4 years)
Average %:		3.86% 5.20% ^b^ 3 years	19.39% 23.89% ^b^ 3 years

* Average % was calculated using recidivism values from known RTC’s (LEGACY), and sexual offenders. ^a^ Considered true allegations by author and counted as recidivism in their study. ^b^ Average recidivism rate including authors count of “true allegations”.

## Data Availability

Data from this study is available upon request through the corresponding author.

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
