# Peer review of "Effectiveness of Residential Treatment for Juveniles with Problematic Sexual Behavior: A Systematic Review"

_ijerph, 2022, doi:10.3390/ijerph192315625_

Round 1

Reviewer 1 Report

Howey et al are trying to highlight the effectiveness of residential treatment for juveniles with problematics sexual behavior by performing a systematic review on recidivism rates. I find the manuscript is very interesting although only six articles have fulfilled the inclusion criteria. the number of articles is very. Nonetheless, I have a few suggestions as follows:

Methodology:

I’d provide the date when the search is terminated.

There are 20 databases involved thus,  I’d provide a table showing the number of retrieved records for each database.

I’d provide the information on the appraisal of the risk of bias (low or high risk).

Have the team ever encountered the inaccessible full texts during their search? If they have, then what is their strategy?

I’d state only English full text resources are accepted for the review.

Author Response

Reviewer #1

We appreciate your willingness to review our manuscript and reconsider the revised manuscript. The insights provided by yourself were very helpful. The revised manuscript is, we believe, stronger and more sound from this effort.

1. I’d provide the date when the search is terminated.

Our Response: We agree and have included the date the search was terminated. This can be located on page 3 of the revised manuscript.

  1. There are 20 databases involved thus,  I’d provide a table showing the number of retrieved records for each database.

Our Response: We appreciate the attention to detail from Reviewer 1; however, we respectfully chose to not follow this recommendation for the following reasons. First, PRISMA guidelines do not ask for such differentiation and, second, a single article can be referenced in multiple databases obscuring the value of such differentiation.

  1. I’d provide the information on the appraisal of the risk of bias (low or high risk).

Our Response: Reviewer 1 brings up a fair point. However, because we didn’t do across-study data synthesis and analyses, all bias is within each study. We’ve added a note to the readers of this point and limitation in the discussion section. This can be located on page 13.

  1. Have the team ever encountered the inaccessible full texts during their search? If they have, then what is their strategy?

Our Response: In all cases, the full text was secured. We have added this in the text. This can be located on page 7.  

  1. I’d state only English full text resources are accepted for the review.

Our Response: We agree and have included this in the methods section. This can be located on page 3.

Reviewer 2 Report

Thank you for giving me the opportunity to review the manuscript entitled "Effectiveness of residential treatment for juveniles with problematic sexual behavior: a systematic review".

This manuscript aims to conduct a systematic review to explore the effectiveness of residential treatment facilities for juveniles with problematic sexual behaviors using recidivism rates as the indicator of effectiveness.

In this study, the authors conduct a systematic review following the PRISMA methodology. You will find my comments below.

The abstract must be rewritten and correctly structured.

Methodology.

Reference latest PRISMA recommendations. 

The methodology should be improved. This paper does not specifically follow the PRISMA protocol, it should be improved and adapted to each of the corresponding headings. For example, they mix the content of results with that of methodology. In addition, the PRISMA diagram should follow the latest recommendations.

There is a very limited number of articles, it is true that the authors point out this difficulty in the limitations, however, in these cases it may be more interesting to modify the type of review work to be carried out, for example a scoping review. 

The search for this review was conducted two years ago. That is a long time for a study of these characteristics, I think it should be a more current search. 

Regarding the descriptive statistical data in section 4.2, this has not been included in the information analysis section of the methodology. This is not usually done in a qualitative systematic review, and if it were a quantitative one it would be a meta-analysis, which is not the case. Statistical data appear in the results and they have not included this in their methodology. 

When talking about the characteristics of the interventions, the studies to which each intervention refers should be indicated. The results are not at all clear to me. I think they should be reorganized. 

I think the discussion is quite correct. It would be interesting to add a section on conclusions. 

Author Response

Reviewer #2

We appreciate your willingness to review our manuscript and reconsider the revised manuscript. The insights provided by yourself were very helpful. The revised manuscript is, we believe, stronger and more sound from this effort.

  1. The abstract must be rewritten and correctly structured.

Our Response: The abstract has been updated to follow the correct structure. It is located on page 1.

  1. Methodology: Reference latest PRISMA recommendations. 

Our Response: We have added in the reference. It can be located on page 3 and in the reference list.

  1. The methodology should be improved. This paper does not specifically follow the PRISMA protocol, it should be improved and adapted to each of the corresponding headings. For example, they mix the content of results with that of methodology. In addition, the PRISMA diagram should follow the latest recommendations.

Our Response: We have re-arranged the format of the manuscript. The updates can be seen on pages 3-9.

  1. There is a very limited number of articles, it is true that the authors point out this difficulty in the limitations, however, in these cases it may be more interesting to modify the type of review work to be carried out, for example a scoping review. 

Our Response: We appreciate Reviewer 2’s appreciation for unique differences between a scoping review and systematic review/meta-analysis; however, given that we started the review as a systematic review/meta-analysis we feel it appropriate to stay with our original methodology. We did however include this recommendation in the limitation section. It can be located on page 13.

  1. The search for this review was conducted two years ago. That is a long time for a study of these characteristics, I think it should be a more current search. 

Our Response: We appreciate Reviewer 2’s recommendation. Because of this recommendation, we expanded our search dates to yesterday’s date and nothing else was found.

  1. Regarding the descriptive statistical data in section 4.2, this has not been included in the information analysis section of the methodology. This is not usually done in a qualitative systematic review, and if it were a quantitative one it would be a meta-analysis, which is not the case. Statistical data appear in the results and they have not included this in their methodology. 

Our Response: Our intention was to conduct a meta-analysis; however, there were not sufficient studies to warrant synthesizing. We do report in our methodology section (section 3.2; page 4) that we coded for participant characteristics. Therefore, we reported it in the results section.

  1. When talking about the characteristics of the interventions, the studies to which each intervention refers should be indicated. The results are not at all clear to me. I think they should be reorganized. 

Our Response: We appreciate the recommendation and feel that Table 2 and the in-text reference to Table 2 accommodates this recommendation. This is located on pages 9 and 10.

  1. I think the discussion is quite correct. It would be interesting to add a section on conclusions. 

Our Response: We agree and have added a Conclusions section. This can be located on page 14.

Round 2

Reviewer 2 Report

The authors have made modifications to their manuscript, but I believe that the final quality of the manuscript has not improved substantially. I find shortcomings in the structure and organisation of the information. 

Instead of talking about previous reviews or meta-analyses, since there are no recent studies, it would be better to include a background section or omit this subtitle. 

Line 11-118, I don't understand this paragraph. They are outcomes, and you are not measuring anything.  

In the same way, table 1 is results, it is not a methodology to carry out the systematic review. 

Do you perform quality appraisal?

What is the basis for data extraction and synthesis?

PRISMA diagram is not the current one. 

There is a lack of bibliographic references in the discussion. 

Author Response

We appreciate your willingness to review our manuscript and reconsider the revised manuscript.

  1. The authors have made modifications to their manuscript, but I believe that the final quality of the manuscript has not improved substantially. I find shortcomings in the structure and organization of the information. 

Our response: We have adjusted the structure and organization of the manuscript. We agree that there are shortcomings to the literature on this topic. However, we believe we followed a rigorous method and have sufficiently answered the question we sought to answer.

  1. Instead of talking about previous reviews or meta-analyses, since there are no recent studies, it would be better to include a background section or omit this subtitle. 

Our response: We agree. The subtitle for this section has been omitted.

  1. Line 11-118, I don't understand this paragraph. They are outcomes, and you are not measuring anything.  

Our response: We agree. This has been removed this paragraph.

  1. In the same way, table 1 is results, it is not a methodology to carry out the systematic review. 

Our response: We agree. Table 1 has been moved to the results section.  

  1. Do you perform quality appraisal?

Our response: We agree. Our quality appraisal was added to the manuscript in the methodology section and the results section.

  1. What is the basis for data extraction and synthesis?

Our response: As is common to systematic reviews, we extracted and synthesized the data available to us from primary studies. Given the limited number of studies, our approach to present data from original studies without submitting them to meta-analytic syntheses seems appropriate. Further, the data we extracted are in line with our primary research questions about evidence on such interventions.

  1. PRISMA diagram is not the current one. 

Our response: This has been updated.  

  1. There is a lack of bibliographic references in the discussion. 

Our response: Two authors reviewed the discussion section and added in relevant bibliographic references.